# Transcriptional Profiling and Molecular Characterization of the *yccT* Mutant Link: A Novel STY1099 Protein with the Peroxide Stress Response and Cell Division of *Salmonella enterica* Serovar Enteritidis

**DOI:** 10.3390/biology8040086

**Published:** 2019-11-13

**Authors:** Sinisa Vidovic, Xiaoying Liu, Ran An, Kristelle M. Mendoza, Juan E. Abrahante, Anup K. Johny, Kent M. Reed

**Affiliations:** 1Department of Veterinary and Biomedical Sciences, University of Minnesota, Saint Paul, MN 55108, USA; liux2725@umn.edu (X.L.); ran.an@uky.edu (R.A.); mendo008@umn.edu (K.M.M.); reedx054@umn.edu (K.M.R.); 2University of Minnesota Informatics Institute, University of Minnesota, Minneapolis, MN 55455, USA; abrah023@umn.edu; 3Department of Animal Science, University of Minnesota, Saint Paul, MN 55108, USA; anupjohn@umn.edu

**Keywords:** *Salmonella enterica* serovar Enteritidis, STY1099 protein, peroxide stress response, nitrate reductase, cell division

## Abstract

Uncharacterized protein STY1099, encoded by the *yccT* gene, was previously identified as the most altered (i.e., upregulated) protein among the ZnO nanoparticle (NP) stimulon of *Salmonella*
*enterica* serovar Enteritidis. Here we combined various stress response-related assays with functional genetics, global transcriptomic and proteomic analyses to characterize the *yccT* gene and its STY1099 product. Exposure of *S*. *enterica* Enteritidis to H_2_O_2_ (i.e., hydrogen peroxide) resulted in a significant (*p* < 0.0001) upregulation of the *yccT* gene, whereas exposure to paraquat (i.e., superoxide) did not alter the expression of the *yccT* gene. The ∆*yccT* mutant of *S. enterica* Enteritidis exposed to 0.75 mM H_2_O_2_, showed significantly reduced (*p* < 0.05) viability compared to the wild type strain. Further, comparative transcriptome analyses supported by Co-immunoprecipitation (Co-IP) assay revealed that STY1099 protein plays a role in redox homeostasis during the peroxide stress assault via involvement in the processes of respiratory nitrate reductase, oxidoreductase activities, cellular uptake and stress response. In addition, we found that the STY1099 protein has the monopolar subcellular location and that it interacts with key cell division proteins, MinD, and FtsH, as well as with a rod shape-determining protein MerB.

## 1. Introduction

Non-typhoidal *Salmonella* (NTS) are zoonotic pathogens of global health importance [1]. The lifestyle of NTS includes frequent multi-host transmission events and short- or relatively long-term survival in the external environment outside animal hosts [2]. To respond to these changing conditions, NTS has acquired a variety of adaptive stress response mechanisms that ensure long-term survival of the pathogen in harsh environments [3,4]. Response to oxidative stress is considered a critical adaptive mechanism for NTS, both within the host [5] and outside the primary habitat of this pathogen [6].

Generally, *Salmonella enterica* possesses two distinct oxidative stress-response systems: (i) a peroxide stress-response system and (ii) a superoxide stress-response system [7]. OxyR, a 34-kDa protein composed of an N-terminal DNA-binding motif and a C-terminal regulatory domain, is the major transcriptional regulator for the expression of the peroxide stress-response genes [8]. OxyR controls the expression of genes encoding enzymes that degrade peroxide; catalase (KatG) and alkyl hydroperoxide-NADPH oxidoreductase (AhpCF), proteins involved in DNA protection (DpS), redox balance (GorA, GrxA and TrxC), as well as repressors of iron transport (Fur) [9]. In addition to direct transcriptional control, OxyR controls numerous genes indirectly, via the synthesis of the small regulatory RNA (*oxyS*) by affecting mRNA stability and translational efficiency [10].

A two-regulatory component system, superoxide radical response (SoxRS) is essential for superoxide stress response [11]. Exposure of NTS to a superoxide-generating agent (e.g., paraquat) leads to synthesis of SoxR, a 17-kDa protein of the MerR family of transcriptional activators, which further initiates expression of the *soxS* gene [12]. Upregulation of the *soxS* results in the activation of the *soxRS* regulon, which includes an efflux system (AcrAB), the manganese-containing superoxide dismutase (SodA), a DNA repair system (endonuclease IV, Nfo), iron uptake (Fur) and electron transport (FldA and FldB).

The oxidative stress regulon commonly overlaps with other stress regulons, such as osmotically inducible genes (*chaA* and *nrdHIEF*), heat stress-response genes (*dnaJ*, *dnaK*, *htpX*, *clpB*, *hslO*, *hslU* and *hslV*), further indicating a strong pleiotropic effect of the oxidative stress response on bacterial physiology [13]. Exploring the antimicrobial effect of zinc oxide nanoparticles (ZnO NPs), specifically their ability to prevent *S*. *enterica* serovar Enteritidis from forming biofilms, we discovered a group of stress response proteins that showed significant upregulation with exposure to ZnO NPs [14]. This group of stress-response proteins consisted primarily of chaperones, proteases governing cell wall, membrane and envelope biogenesis (HtrA, DegP, ClpC, TolB, AotJ, GroEL, and CspC), as well as a hypothetical protein (STY1099). Among this ZnO response stimulon, STY1099 exhibited the most significant upregulation (16-fold), indicating a possible role in protecting *S*. Enteritidis from reactive oxygen species generated by ZnO NPs.

In this study, we employed functional genetics and proteomics as well as global transcriptomic approaches to characterize the *yccT* gene that encodes STY1099 protein, a novel prokaryotic stress response protein, using *S. enterica* subs. *enterica* serovar Enteritidis as a model organism.

## 2. Materials and Methods

### 2.1. Bacterial Strains, Plasmids and Growth Conditions

*Escherichia coli* DH5α was used as host for the recombinant plasmid pTre99A, while *Salmonella enterica* subs. *enterica* serovar Enteritidis ATCC 13076 strain served as the wild type. Plasmid pKD3 was used as a template for amplification of the Cm resistance cassette and the pTre99A was used as an expression vector. Plasmids pKD46 and pCP20 were used during the Red Lambda procedure. Growth media was supplemented with ampicillin (100 μg/mL), chloramphenicol (30 μg/mL) and arabinose 10 mM (Sigma Chemical Co., St. Louis, MO, USA) for maintenance of plasmids and selection of bacterial strains, as required.

### 2.2. Gene Expression Assay

Overnight cultures of the wild type *S*. *enterica* Enteritidis strain were diluted 1/100 in 100 mL of Luria–Bertani (LB) medium and grown at 37 °C with constant shaking at 190 rpm to optical density at 600 nm of 0.5. This mid-exponential growth phase culture was exposed to 3 mM of H_2_O_2_. Cultures were additionally incubated for 60 min and harvested by centrifugation. Total RNAs were extracted using the RNeasy Mini kit (Qiagen), following the manufacturer’s instructions. Synthesis of complementary DNA (cDNA) was carried out using iScriptTM Reverse Transcription (Bio-Rad Laboratories, Inc., Hercules, CA, USA). Quantitative PCR was performed using Power SYBR green master mix kit (Applied Biosystems, Foster, CA, USA). The *rtcR* gene was selected as an internal reference control and the data were reported as the fold change relative to levels in the susceptible strains using the comparative C_T_ method [14].

### 2.3. Construction of ∆yccT, ∆napB, ∆gutM and ∆ahpCF Salmonella enterica Enteritidis Strains

Construction of chromosomal deletions was performed using the Red Lambda recombination system as previously described [15]. Briefly, the chloramphenicol resistance cassette, *cat*, flanked by Flp recognition sites, was amplified using the pKD3 plasmid as the DNA template. All primers used for the construction of mutants and for the complementation study are listed in Table 1. Amplified *cat* cassettes were used to transform the wild type strain harboring the Red recombination plasmid, pKD46. Introduction of desirable mutations was verified by PCR and DNA Sanger sequencing. The deletions were transferred to strains by P22 transduction. To excise the *cat* cassette, a temperature sensitive Flp recombinase-expressing vector, pCP20, was introduced via electroporation. The pCP20 plasmid was further cured by growing the mutants at an elevated temperature (42 °C). All mutants were verified with PCR and sequencing and used for later functional analysis.

### 2.4. Oxidative Stress Killing Assay

The oxidative killing assay was performed using overnight cultures of the wild type *S. enterica* Enteritidis ATCC 13076 and its ∆*yccT*, ∆*napB*, ∆*gutM* and ∆*ahpCF* mutant strains grown in LB at 37 °C with constant shaking at 180 rpm. Seed cultures were diluted 1/100 in 100 mL freshly prepared LB and grown to optical density at 600 nm of 0.4. Each wild type and mutant culture (∆*yccT*, ∆*napB*, ∆*gutM* and ∆*ahpCF*) was exposed at mid-exponential growth phase to H_2_O_2_ in a final concentration of 3 mM. Oxidative stress was maintained for 60 min under the same temperature and shaking conditions. Viable cell counts were carried out by 10-fold dilutions in 0.9% NaCl at time zero (before H_2_O_2_ exposure) and then at every 15 min of incubation. Aliquots of 0.1 mL were plated on LB agar in triplicate and then incubated at 37 °C for 24 h.

### 2.5. Complementation Study

A DNA fragment containing the *yccT* gene, plus 150 bp upstream and downstream was amplified using the genomic DNA of *S. enterica* Enteritidis ATTC 13076 as template. Plasmid pTrc99A was digested with *Sma*I and *Sal*I restriction enzymes, and the 975 bp product containing an integral *yccT* gene was cloned into pTrc99A. The recombinant plasmid pTrc99A-*yccT* was introduced into ∆*yccT S. enterica* Enteritidis ATCC 13076 strain, and transformants were selected on LB agar containing ampicillin. As control for the complemented ∆*yccT S. enterica* Enteritidis ATCC 13076 strain, the wild type *S. enterica* Enteritidis ATTC 13076 and ∆*yccT S. enterica* Enteritidis ATCC 13076 strains were transformed with an empty pTrc99A vector.

### 2.6. Preparation of Samples for RNA Extraction

Untreated and 30 min treated cultures (10 mL volume) of the wild type and its three mutants were centrifuged at 4000 rpm for 5 min. The resultant cell pellets were washed two times followed by RNA extraction using the RNeasy Mini kit (Qiagen) following the manufacturer’s instructions. Sample quality was assessed using capillary electrophoresis (e.g., Agilent BioAnalyzer 2100), generating an RNA integrity number (RIN). A RIN of eight or greater was required to pass initial QC.

### 2.7. RNA-Seq Analysis

After quality control, RNA samples were converted to Illumina sequencing libraries using Illumina’s TruSeq Stranded Total RNA Library Prep Human/Mouse/Rat Sample Preparation Kit (Cat. # 20020597) at the University of Minnesota Genomics Center. Approximately 500 nanograms of total RNA was rRNA depleted using sequence-specific Ribozero capture probes. The mRNA was then fragmented and reverse transcribed into cDNA. The cDNA fragments were blunt-ended and ligated to indexed (barcoded) adaptors and amplified using 15 cycles of PCR. Final library size distribution was validated using capillary electrophoresis and quantified using fluorimetry (PicoGreen). Indexed libraries were then normalized and pooled in an equimolar fashion. Truseq libraries were hybridized to a single read flow cell and individual fragments were clonally amplified by bridge amplification on the Illumina cBot. Once clustering was complete, the flow cell was loaded on the HiSeq 2500 and sequenced using Illumina’s SBS chemistry. Base call (bcl) files for each cycle of sequencing were generated by Illumina Real Time Analysis (RTA) software. Primary analysis and index de-multiplexing were performed using Illumina’s bcl2fastq v2.20.0.422. The end result of the bcl2fastq workflow resulted in creation of de-multiplexed FASTQ files (i.e., raw sequence data). Quality control on raw sequence data for each sample was performed with FastQC. Read mapping was performed via Hisat2 (v2.0.2) using the *Salmonella enterica* Enteritidis genome P125109 as reference. Gene quantification was done via Feature Counts for raw read counts. Differentially expressed genes were identified using the edgeR (negative binomial) feature in CLCGWB (Qiagen, Valencia, CA, USA), followed by filtration based on a minimum 2× absolute fold change and false discovery rate (FDR) corrected *p* < 0.05.

### 2.8. Real-Time PCR

The RNA Seq data were validated by quantitative real-time PCR (qRT-PCR). Synthesis of cDNA was carried out using iScript^TM^ Reverse Transcription (Bio-Rad Laboratories, Inc. Hercules, CA, USA). qRT-PCR was performed on a MiniOpticon^TM^ Real-Time PCR Detection System (Bio-Rad Laboratories, Hercules, CA, USA) with iQTM SYBR^®^ Green Supermix kit (Bio-Rad Laboratories, Hercules, CA, USA). Gene *rtcR*, encoding for transcriptional regulator protein RtcR, was selected as an internal reference gene. The expression of *rtcR* in the wild type and its isogenic *yccT* mutant was not affected by H_2_O_2_ treatment or *yccT* mutation. Primers of seven genes (*ego*, *pduA*, *ydeZ*, *ydeV*, *yneC*, *rplP* and *rplV*) were designed to generate internal fragments ranging from 91 bp to 129 bp in size (Appendix A).

### 2.9. Subcellular Localization of STY1099

Subcellular localization of the STY1099-fused with green fluorescent protein (GFP) was carried out in the transformed *S. enterica* Enteritidis ATCC 13076 wild type strain by inverted TiE deconvolution microscope. Briefly, the *yccT-egfp fusion* gene was amplified using the genome of *S. enterica* serovar Enteritidis ATCC 13076 and egfp plasmid (AddGene, Watertown, MA, USA) as the DNA template. The primers for *yccT* PCR amplification were 5′–GGAATTCCATATGAAAACCGGCGCGCTAGCCACCTT–3′ and 5′–TCCTCGCCCTTGCTCACCATGGATCCAGAGGGCGGCTGTTTTTCCGC–3′and the primers for *egfp* PCR amplification were 5′–GCGGAAAAACAGCCGCCCTCTGGATCCATGGTGAGCAAGGGCGAGGA–3′ and 5′–GGAATTCCATATGAAAACCGGCGCGCTAGCCACCTT–3′. Fusion of *yccT* and *egfp* was carried using amplified *yccT* and *egfp* as DNA templets and primers 5′–GGAATTCCATATGAAAACCGGCGCGCTAGCCACCTT–3′ and 5′–GGAATTCCATATGAAAACCGGCGCGCTAGCCACCTT–3′. The amplified *yccT-egfp* gene was digested with NdeI and XholI and ligated into the pET42b vector. An egfp gene intended for non-fusion was amplified with 5′–GGAATTCCATATGGTGAGCAAGGGCGAGGA–3′ and 5′–CCGCTCGAGTTACTTGTACAGCTCGTCCA–3′. The amplified egfp gene was digested with NdeI and XhoIl and followed ligation into the pET4b vector. After plasmid isolation and sequence verification, the pET42b::*yccT-egfp* and pET42::egfp plasmids were used to transform *S. enterica* Enteritidis ATCC 13076 wild type. Concentration of 0.5 mM of isopropyl ß-D-1-thiogalactopyranoside (IPTG) was used to induce expression of the *yccT*EGFP and EGFP, respectively. Transformed *S. enterica* Enteritidis cells were spotted on slides with ProLong^®^ Diamond Antifade Mountant (4′6-diamidino-2-phenylindole) and imaged. Images were acquired in a Nikon TiE microscope equipped with a confocal A1R scan head. All images were captured with a 100× Apo TIRF 1.49 NA objective. GFP fluorescence was excited with a 488 nm laser and fluorescence collected through a 500–550 nm emission filter. The confocal aperture was set to 16.6 µm (0.3 AU) and the voxel size was 31 × 31 × 100 nm (XYZ). All acquisition parameters were kept constant between experimental and control samples. The images were processed with 3D automatic deconvolution (Nikon Elements software, 20 iterations).

### 2.10. Co-Immunoprecipitation of STY1099

Co-immunoprecipitation (Co-IP) assay was carried out at 4 °C unless otherwise indicated, using *E. coli* 21 and *S. enterica* Enteritidis ATCC 13076 strains. Briefly, the fusion *yccT*-flag gene was amplified using the genome of *S. enterica* Enteritidis ATCC 13076 as the DNA template. The primers for PCR amplification were 5′–GGA ATTCCATATGAAAACCGGCGCGCTAGCCACCTT–3′ and 5′–CCGCTCGAGTCACTTGT CGTCATCGTCTTTGTAGTCAGAGGGCGGCTGTTTTTCCGCCCATT–3′. The amplified *yccT*-flag gene was digested with NdeI and XholI and ligated into the pET42b vector. Two plasmids, pET42::*yccT* and pET42b empty vector, were used to transform *E. coli*, respectively. *E. coli* pET42b was used as negative controls. Expression of *yccT* was induced using 0.5 mM of IPTG. The expression of yccT was confirmed by performing Western blot. To pull down the STY1099-flag tagged protein, the lysates (*E. coli* served as negative control, or mixture of *E. coli* and *S. enterica* Enteritidis) were incubated with anti-FLAG M2 agarose (Sigma) at 4 °C overnight on an end-over-end rotator. After centrifugation the beads were washed with lysis buffer three times followed by proteins elution. The samples of eluted proteins were loaded onto a 10% SDS-PAGE gel. The protein bands were visualized by Coomassie Brilliant Blue R-250 (Thermo Fisher Scientific, Waltham, MA, USA) and excised for protein identification. After in-gel trypsin digestion the peptide mixture was analyzed by capillary liquid chromatography-mass spectrometry (LC-MS) on an Eksigent 1D plus LC with a MicroAS autosampler (Dublin, Ireland) online with an Orbitrap Velos MS system (Thermo Fisher Scientific, Waltham, MA, USA) as previously described [16].

### 2.11. Experimental Replications and Bioinformatics

Data from the gene expression, oxidative killing assays, global transcriptomics and RT-qPCR represent the average of three biological replicates. Kinetic data were analyzed by CoStat version 6.4 software (Co-Hort Software, Monterey, CA, USA) using the homogeneity of linear regression slopes method to test for significant (*p* < 0.05) differences. Gene expression data were analyzed by *t*-tests. Nucleotide sequence translation was carried out using EMBOSS Transeq [17] (the European Molecular Biology Laboratory—European Bioinformatics Institute; Hinxton, Cambridge, United Kingdom). The neighbor-joining algorithm [18] was used to generate the phylogenetic tree. The protein sequences were aligned using Clustal Omega [19] and colored using the “Percentage Identity” color-scheme in Jalview [20]. The STY1099 protein binding site and tertiary structure predictions were determined using RaptorX algorithms [21]. The gene ontology (GO) analysis was conducted using the Database for Annotation, Visualization and Integrated Discovery (DAVID) [22]. Signal peptide was determined using the SignalP 5.0 software version [23].

## 3. Results

### 3.1. Phylogeny, Conserved Domains and Structural Characteristics of STY1099 Protein

Phylogeny of the *yccT* gene sequence greatly resembled taxonomic relationships of the tested species (Figure 1A). Four different *Salmonella* serovars formed a distinct cluster, while *Shigella flexneri*, a close relative of the *Escherichia coli* species, clustered together with *E. coli* (Figure 1A)*. Vibrio cholerae* (*Vibrionaceae*), *Haemophilus influenzae* (*Pasteurellaceae*) and *Serratia* spp. (*Enterobacteriaceae*) were joined to the main tree basally, indicating a distant phylogeny of their *yccT* sequences (Figure 1A). 

The overall average pairwise distance among the three families, *Enterobacteriaceae*, *Vibrionaceae* and *Pasteurellaceae*, was 0.437, indicating any two bacterial strains from this group would diverge, on average, 43.7% from each other based on the *yccT* gene sequence. Alignment of the STY1099 protein sequence revealed numerous conserved domains, including amino acid positions, 1, 2, 21, 34, 38, 42, 62, 83, 90, 103, 135, 137, 155, 156, 162, 200, 203, 206, 210, 211, 214 and 218 (Figure 1B). The STY1099 protein sequences, from this diverged group of bacteria, exhibited common binding sites with high pocket multiplicity (PM) values for SO_4_ (PM 20), Zn^+^ (PM 20), Na^+^ (PM 9) and HO (PM 9). Interestingly, the first amino acid binding site for each ligand, SO_4_ (L42), Na^+^ (N156), Zn (Q62) and HO (L34) was completely conserved among the tested species (Figure 1B), suggesting a ubiquitous physiological role of this protein. Based on the *S. enterica* Enteritidis protein sequence the STY1099 protein contains two primary domains; domain 1 (*p* = 6.07 × 10^−3^; from 1 to 158 aa) and domain 2 (*p* = 4.14 × 10^−2^; from 159 to 220 aa), forming four α-helixes and three antiparallel β-pleated sheets (Figure 1C). Also, the same protein sequence, with a high likelihood (0.9497), contains a Sec/SPI “standard” secretory signal peptide with a cleavage site located between positions 20 and 21 and VFA/TT amino acid sequences (probability 0.9324).

### 3.2. The yccT Gene Is Highly Inducible upon Exposure to Hydrogen Peroxide, But Not to Parquet

In our previous study we carried out a STRING analysis to identify an interactome of protein STY1099 [23]. This analysis with a high confidence score (confidence score, 0.70) showed that the STY1099 interacts with several proteins in the proteome of *S. enterica* Enteritidis including NapD (an assembly protein for periplasmic nitrate reductase) and GutM (a DNA-binding transcriptional activator). To determine whether or not reactive oxygen species caused upregulation of STY1099 and its interactome (NapD and GutM proteins), we tested the expression of genes, *yccT* (STY1099), *gutM* (GutM), *napD* (NapD) and two reference genes *oxyR* and *sodA*, using peroxide (H_2_O_2_) and superoxide (paraquat) as oxidative agents. No significant changes in expression of *yccT*, *napD* and *gutM* were observed during treatment of *S. enterica* Enteritidis with paraquat (Appendix A). In sharp contrast, exposure of *S. enterica* Enteritidis to H_2_O_2_ resulted in a significant upregulation of the *yccT* gene. Figure 2 shows the expression dynamics of *yccT*, *gutM* and *napD* alongside *oxyR* and *sodA* in the presence of 3 mM H_2_O_2,_ and in controls over 60 min.

Most notably, the expression of *yccT* exhibited a profound upregulation with the H_2_O_2_ treatment. Significant upregulation (*p* < 0.0001) of *yccT* was observed at 15 min (11.68-fold), 30 min (18.71-fold) and 45 min (9.95-fold) of the peroxide treatment. Induction of *yccT* during peroxide treatment was greater than that of *oxyR*, which is a regulatory hallmark of the peroxide stress response in *S. enterica*. Interestingly, another positive control, *sodA*, showed a massive change in expression in the presence of H_2_O_2_, resulting in three- and six-times higher expression compared to those of *yccT* and *oxyR*, respectively, during 30 and 45 min treatments (Figure 2). A reason of the massive expression of *sodA* most likely lies in the fact that this gene encodes an enzyme, SodA, which catalyzes an oxidative agent, whereas *oxyR* encodes a protein that regulates expression of other oxidative stress response genes. In other words, *sodA* must be expressed at much higher levels compared to that of *oxyR*, as these two proteins have quite different modes of action combating an oxidative assault. Based on the expression level, the *yccT* gene is positioned between these two oxidative stress response genes, *oxyR* and *sodA*. This may indicate that *yccT* has a different mode of action during oxidative stress response of *S. enterica* compared to those of *oxyR* and *sodA*. Both *yccT* and *oxyR* exhibited the same pattern of gene expression, a significant increase at 15 min, followed by highest expression at 30 min and a return to basal levels after 60 min (Figure 2). Two other genes, *napD* and *gutM*, did not show significant induction during peroxide treatment.

To evaluate the sensitivity of *yccT* to peroxide stress, we treated *S. enterica* Enteritidis with three different concentrations (0.75 mM, 1 mM and 3 mM) of H_2_O_2_ over the same period of time. Data show *yccT* significantly upregulated (*p* < 0.005) at concentrations of 0.75- and 1-mM after 15 min treatment followed by a sharp decrease in the gene expression (Figure 3). The efficiency of RT-qPCR primers are presented in Appendix A and the expression data for the reference gene *rtcR* are provided in Appendix A.

Most notably, the expression of *yccT* exhibited a profound upregulation with the H_2_O_2_ treatment. Significant upregulation (*p* < 0.0001) of *yccT* was observed at 15 min (11.68-fold), 30 min (18.71-fold) and 45 min (9.95-fold) of the peroxide treatment. Induction of *yccT* during peroxide treatment was greater than that of *oxyR*, which is a regulatory hallmark of the peroxide stress response in *S. enterica*. Both *yccT* and *oxyR* exhibited the same pattern of gene expression, a significant increase at 15 min, followed by highest expression at 30 min and a returned to basal levels after 60 min (Figure 2). Two other genes, *napD* and *gutM*, did not show significant induction during peroxide treatment.

### 3.3. The ∆yccT Mutant Exhibits Increased Sensitivity to Hydrogen Peroxide

After demonstrating *yccT* induction by peroxide, we investigated whether *yccT* plays a role in hydrogen peroxide tolerance of *S*. *enterica* Enteritidis. Two additional genes, *gutM* and *napD*, that encode interactome proteins of STY1099 were also included in oxidative killing assays. As a reference strain with known susceptibility to oxidative stress, the *ahpCF* mutant strain of *S. enterica* Enteritidis was included in this assay too. Sensitivity of the ∆*yccT*, ∆*gutM*, ∆*napD* and ∆*ahpCF* mutants to oxidative stress was determined by exposing exponentially grown cultures of these mutants and their wild type strain to 0.75 mM of H_2_O_2_ over 60 min. Concentrations of H_2_O_2_ higher than 0.75 mM affected the wild type strain too, so for the oxidative killing assay concentration of 0.75 mM was selected. Homogeneity of linear regression slope analysis revealed a statistically significant difference (*p* < 0.05) in the viability between the wild type and *yccT* mutant, suggesting that the *yccT* gene plays a role in oxidative tolerance of *S. enterica* Enteritidis (Figure 4A).

Two other mutants, ∆*gutM* and ∆*napD*, showed reduced viability, albeit less pronounced compared to the ∆*yccT* mutant. The ∆*ahpCF* mutant showed the highest level of susceptibility to H_2_O_2_ during the first 40 min of assay. After this period, the ∆*ahpCF* mutant showed an increased survivability, which resulted in a higher survival rate compared to that of the ∆*yccT* mutant (Figure 4A). During hydrogen peroxide treatment, the ∆*yccT* mutant complemented with the pTre99A-*yccT* vector had increased viability during the peroxide treatment compared to the ∆*yccT* mutant (Figure 4B).

### 3.4. Subcellular Localization of STY1099

As STY1099 protein contains a conserved domain of unknown function 2057 (DUF2057), which occurs only in prokaryotes, we aimed to determine subcellular location of this prokaryotic protein. STY1099 was fused with green fluorescent protein (GFP) and expressed in *S. enterica* Enteritidis cells by a lac IPTG (isopropyl ß-D-1-thiogalactopyranoside) inducible promoter. Inverted TiE microscopy coupled with confocal scanning option showed strong and consistent fluorescence signal in transformed *S. enterica* Enteritidis cells, clearly indicating monopolar location of STY1099 protein (Figure 5).

These data suggest that STY1099 protein has a specific function in bacterial cells that may correspond to cell motility (flagella), cell division or other cellular function that requires monopolar location of the studied protein.

### 3.5. Global Transcriptomics of the S. enterica Enteritidis ∆yccT Mutant Strain Reveal the STY1099 Regulon

To determine the effect of *yccT* deletion on the oxidative stimulon of *S. enterica* Enteritidis, and specifically to identify proteins affected by the STY1099 protein, we carried out global transcriptome (RNAseq) analyses using the wild type and the ∆*yccT* mutant strains in combination with the oxidative stress assay. Alterations in gene expression were studied by comparing the transcriptomes with a minimum 2× absolute fold change and false discovery rate (FDR) corrected *p* < 0.05 used as cutoffs for significance.

In total, 2051 genes were significantly differently expressed (DE) by H_2_O_2_ in the wild type strain, and 1712 DE genes were observed in the ∆*yccT* mutant (Appendix A). Of the 1712 DE genes, 1453 were shared between the comparisons, while 259 were unique to the *yccT* mutant and 598 unique to the wild type treatment group. There were 22 genes exhibiting statistically significant changes in expression between the wild type and *yccT* mutant during the H_2_O_2_ treatment (Table 2). Out of the 22 genes, 13 genes (*narGHIJK*, *hycDF*, *dmsA3*, *rrmJ*, SEN0541, SEN1163, SEN1249 and SEN3184) were unique to the oxidative response, and nine genes (*citEFT*, *kdgT*, *nirC*, *yccT*, SEN0167, SEN0271 and SEN0992) of the oxidative stimulon were shared with the core group of the *yccT* affected genes (Appendix A). When exposed to oxidative stress, the ∆*yccT* mutant showed significant upregulation (*p* < 0.05) of the citrate metabolism (*citEFT*), oxidoreductase activity/electron transport (*hycDF*, *dmsA3*, SEN1249/3184), rRNA processing/stress response (SEN0992, *rrmJ*) and gluconate transmembrane transport (*kdgT*) (Table 2). The ∆*yccT* mutant significantly downregulated (*p* < 0.05) nitrate metabolism (*narJHIGK* and *nirC*) (Table 2). Also, affected by oxidative stress in the ∆*yccT* mutant was a group of genes that encodes proteins of unknown function (SEN0167, SEN0271, SEN1163 and SEN0541).

To investigate how the *yccT* gene deletion specifically affected the physiology of growing cells, we compared the transcriptomes of exponentially growing culture of the *yccT* mutant to that of the wild type strain. Global transcriptomic analyses identified 352 (343 unique genes) differentially expressed genes in the no treatment controls; 137 genes significantly upregulated and 215 genes downregulated (Appendix A).

The most pronounced changes in biological processes occurred via downregulation of genes involved in macromolecular biosynthetic processes, translation, cell motility and inorganic cation transmembrane transport (Figure 6). Among this large group of 215 genes, the most profound gene expression alterations were exhibited by a group of genes associated with electron transport via the process of nitrate reduction, including *napA* (−22.95; FDR *p* = 0.0), *napB* (−34.04; FDR *p* = 7.92 × 10^−22^), *napC* (−31.42; FDR *p* = 1.88 × 10^−19^), *napD* (−16.35; FDR *p* = 8.76 × 10^−17^), *napF* (−17.48; FDR *p* = 1.87 × 10^−18^), *napG* (−29.60; FDR *p* = 5.58 × 10^−25^) and *napH* (−27.25; FDR *p* = 6.36 × 10^−21^).

Upregulated biological processes of the ∆*yccT* mutant included organo-nitrogen compound metabolic processes, organic hydroxyl and primary amino compound catabolic processes, cellular amine metabolic and branched-chain amino acid biosynthetic processes (Figure 6). This group of upregulated genes had less pronounced expression changes, compared to the downregulated genes. Largest upregulation was observed in genes involved in propanediol metabolism, including *pduA* (17.25; FDR *p* = 8.72 × 10^−36^), *pudB* (12.69; FDR *p* = 7.20 × 10^−31^), *pudD* (10.70; FDR *p* = 1.48 × 10^−32^), and *pudC* (9.77; FDR *p* = 3.43 × 10^−27^).

To validate the RNA-seq data, the expression of seven randomly chosen genes was validated by quantitative real-time PCR (qRT-PCR). The qRT-PCR analysis confirmed the same pattern of expression, including five upregulated genes (e.g., *ego*, *pduA*, *ydeZ*, *ydeV* and *yneC*) and two downregulated genes (e.g., *rplP* and *rplV*) (Figure 7).

### 3.6. Determination of STY1099 Interactome

To identify proteins that are associated with the STY1099 protein, we carried out a Co-IP assay. Using STY1099-flag protein as a bait and proteome of *S*. *enterica* Enteritidis as a pray, we identified 36 proteins that were associated with the STY1099 protein (Table 3). Validation of the Co-IP assay were performed by Western blot, and SDS-PAGE analyses (Figure 8, Appendix A).

This group of STY1099-associated proteins consisted of a group of cell division and shape proteins, including septum site-determining protein MinD, cell division protein FtsH and rod shape-determining protein MreB, indicating that the STY1099 protein may play a role in cell division and determination of cell shape. Besides association with this functionally very specific group of proteins, the STY1099 protein showed interaction with a group of proteins involved in oxidoreductase activity. Proteins involved in oxidoreductase activities perform various biological processes, including lipid metabolism (e.g., multifunctional fatty acid oxidation complex, FadB), amino acid metabolism (e.g., aminoacyl-histidine dipeptidase, PepD), carbohydrate metabolism (e.g., bifunctional acetaldehyde-CoA/alcohol dehydrogenase, AdhE), and electron transport (e.g., cytochrome d terminal oxidase subunit 1) (Table 3). The STY1099 protein interacted with a group of ribosomal and DNA binding proteins involved in transcription and translation as well as several stress response proteins associated with heat stress (e.g., DnaK protein), DNA damage (e.g., DNA-binding ATP-dependent proteases La), osmotic stress (e.g., two-component response regulator OmpR) and extracytoplasmic stress (e.g., ATP-dependent protease ATP-binding subunit ClpX) (Table 3). The STY1099 protein interacted also with two proteins, maltose/maltodextrin transporter ATP-binding protein and oligopeptide ABC transporter ATP-binding protein OppF, involved in cellular uptake (Table 3).

## 4. Discussion

Our previous study identified a protein of unknown function STY1099 (i.e., encoded by *yccT* gene) as the most abundant among a small group of proteins associated with the ZnO NPs stimulon [24]. It is important to emphasize that ZnO NPs affect cells via two distinct pathways: Zn toxicity and oxidative stress [25,26]. The Zn toxicity-mediated pathway is primarily based on inhibition of the major enzymes involved in the tricarboxylic acid cycle [25], glycolysis [27] and the capsule biosynthesis [28]. It has been postulated that lethality of ZnO NPs is also associated with the production of increased levels of reactive oxygen species, mainly hydroxyl (OH) radicals [26], which further interfere with phospholipids, lipopolysaccharides and lipoproteins, leading to extensive cell wall disorganization of Gram-negative bacteria [29]. To distinguish this dual effect of ZnO NPs and, more importantly, to pinpoint a stimulus-causing upregulation of STY1099 in *S. enterica* Enteritidis exposed to ZnO NPs, we measured expression of the *yccT* gene during oxidative treatment. Our data show that *yccT* is significantly overexpressed upon exposure to H_2_O_2_, but not upon exposure to paraquat, indicating a specific role of *yccT* in peroxide homeostasis of this zoonotic pathogen.

We demonstrated that under peroxide stress an *S. enterica* Enteritidis strain lacking *yccT* showed a significant decrease in viability, compared to that of the wild type, indicating that the STY1099 protein likely plays a role in protecting *S. enterica* Enteritidis from the peroxide stress assault. A possible polar effect caused by the *yccT* deletion was ruled out by complementation. Increased viability of the ∆*yccT* mutant complemented with the pTre99A-*yccT* vector clearly showed that intolerance to peroxide stress of the ∆*yccT* mutant is due to the *yccT* deletion.

Transcriptomic analysis of the ∆*yccT* mutant and wild type provided insight into the transcriptome of this mutant in non-stressful conditions. Most notably, a majority (61%) of the altered genes became significantly downregulated including genes associated with macromolecular biosynthetic processes and translation being most common in the ∆*yccT* mutant. Specifically, most significant was downregulation of the *napABCDGHF* operon, which encodes periplasmic nitrate reductase (Nap) proteins that catalyze the transformation of nitrate to nitrite. In prokaryotic organisms, reduction of nitrate to nitrite occurs via three distinct systems, namely periplasmic nitrate reductase (Nap), respiratory nitrate reductase (Nar) and assimilatory nitrate reductase (Nas) [30]. Unlike other nitrate reductase systems, Nap is functionally diverse. It is demonstrated to be involved in dissimilatory nitrate reduction (both denitrification and nitrate reduction to ammonia) [31], maintenance of cellular oxidation–reduction potential (i.e., redox poise) [30] and nitrate scavenging and pathogenicity [32]. Besides nitrate reduction, the ∆*yccT* mutant exhibited a significant downregulation of the *rpsQCS* operon (i.e., from eight- to five-fold), which encodes one of the primary rRNA binding proteins, RpsQ (30S ribosomal protein S17). The RpsQ protein plays a critical role in translational accuracy [33], while RpsC (30S ribosomal protein S3) is involved in mRNA unwinding [34] and RpsS (30S ribosomal protein S19) in ribosomal small subunit assembly [35]. Downregulation of the Nap system and the *rpsQCS* operon in the ∆*yccT* mutant indicates that STY1099 impacts primary nitrate metabolism, cellular oxidation–reduction potential and rRNA biology during exponential growth of *S. enterica* Enteritidis.

Among a group of upregulated genes, the *pduABCDE* operon exhibited the greatest alteration, ranging from 17-fold for the *pduA* gene (propanediol utilization protein A) to 6.4-fold for the *pduE* gene (propanediol dehydratase small subunit). Proteins encoded by the *pduABCDE* operon catalyze the dehydration of 1,2-propanediol to propionaldehyde, which further serves as the carbon and energy source for bacterial growth [36]. The *yccT* mutant also showed significant upregulation of the *yneABC* operon (~ four-fold), which encodes proteins involved in substrate binding, trans-membrane transport, and interconverting aldoses and ketoses [37].

Indeed, our Co-IP assay showed that the STY1099 protein interacts with a group of proteins involved in oxidoreductase activity, electron transport, rRNA biology and cellular uptake, clearly indicating that the STY1099 protein is involved in these cellular processes. Besides these proteins involved in the central metabolic processes, the STY1099 protein interacted with proteins of key importance for cell division (e.g., septum-site determining protein MinD and cell division protein FtsH) and a protein involved in determination of cell shape (e.g., rod shape determining protein MreB). Monopolar location of the STY1099 protein additionally suggests that the STY1099 protein may play a role in cell division and cell shape determination too.

The molecular response of the ∆*yccT* mutant to peroxide stress included downregulation of nitrate reductase (*narJHIGK*, and *nirC*) and upregulation of oxidoreductase activity (*hycDF*, *dmsA3*, SEN1249 and SEN3184) and citrate metabolism (*citEFT*). Interestingly, during exponential growth with no peroxide treatment, the ∆*yccT* mutant exhibited a profound downregulation of nitrate reductase mediated by Nap, while during peroxide treatment nitrate reductase was altered via the respiratory nitrate reductase (Nar) system. Although both Nap and Nar systems belong to dissimilatory nitrate reductases, the Nar system is strictly involved in redox regulation and energy acquisition, whereas the Nap system is functionally diverse [30]. Both nitrate reductases, Nap and Nar, are membrane associated, whereas Nas is made of soluble proteins located in the cytoplasm [38]. We showed that the STY1099 protein only affects membrane associated nitrate reductase, which is further supported by the fact that this novel protein is associated with cellular membrane (i.e., monopolar localization), but not with cytoplasm. In addition, it has been shown with the high likelihood (0.9324) that the STY1099 protein contains a Sec/SPI signal peptide. This secretory signal peptide, transported by the Sec translocon and cleaved by signal peptidase I, is an integral part of periplasmic proteins, which suggests that the novel STY1099 protein belongs to a group of periplasmic proteins. Downregulation of the Nar system in the ∆*yccT* mutant during peroxide treatment indicates that STY1099 plays a role in redox homeostasis. Indeed, upregulation of the genes associated with oxidoreductase activity and citrate metabolism provides a second line of evidence that STY1099 is involved in maintenance of cellular redox balance. In addition to this, interactome of STY1099, defined by the Co-IP assay, showed that the STY1099 protein is associated with a group of proteins involved in oxidoreductase activities. These oxidoreductase proteins are involved in various biological processes (e.g., lipid, amino acid and carbohydrate metabolisms as well as electron transport), clearly indicating that their association with the STY1099 protein is based on their role in oxidoreductase activities rather than involvement in some other biological processes.

Fu and colleagues [38], examining response of *S*. *enterica* Typhimurium to H_2_O_2_ treatment, found that this *Salmonella* serovar significantly induced several proteins associated with DNA repair machinery, including RecA, RecE and GyrI. The current study showed that the STY1099 protein interacted with DNA-binding ATP-dependent protease La, showing that this novel protein is indirectly associated with DNA repair machinery, an important response of *Salmonella enterica* to peroxide stress [38]. Besides DNA-binding proteases La, the STY1099 protein interacted with key protease, ClpX, chaperone, DnaK, and the two-component response regulator, OmpR, which controls a variety of membrane-associated transporters that mediate uptake or efflux of small molecules [39], showing its importance in the stress response physiology.

## 5. Conclusions 

We demonstrated that *yccT* is a highly peroxide-inducible gene, and the ∆*yccT* mutant exhibits peroxide intolerance compared to its wild type strain. Interestingly, the ∆*yccT* mutant showed a tolerance to parquet, indicating that this novel protein is specifically involved in the peroxide stress response of *S. enterica* serovar Enteritidis. Comparative transcriptome analyses supported by Co-IP assay showed that the STY1099 protein is involved in oxidoreductase processes, respiratory nitrate reductase, rRNA biology and stress response. In addition, we showed that the STY1099 protein has monopolar location and that interacts with key cell division proteins MinD and FtsH as well as with a rod shape-determining protein MreB.

## Figures and Tables

**Figure 1 biology-08-00086-f001:**
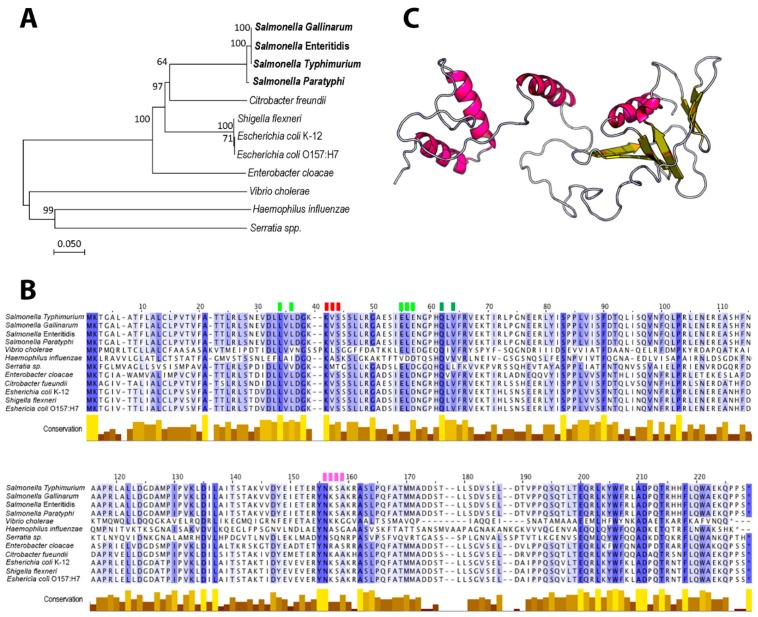
Characterization of the *yccT* gene and its STY1099 protein sequence. (**A**) Phylogeny of the *yccT* gene sequence using seven bacterial species from *Enterobacteriaceae*, *Vibrionaceae*, and *Pasteurellaceae* families. (**B**) Sequence alignment of STY1099 protein from *Salmonella enterica*, *Escherichia coli*, *Shigella flexneri*, *Citrobacter freundi*, *Enterobacter cloacae*, *Vibrio cholerae*, *Haemophilus influenzae* and *Serratia* spp. Color gradient depicts highly conserved sequences (dark blue), moderate conserved (light blue) and no conserved sequences (white). Binding sites for SO_4_ (red), Na^+^ (pink), Zn (dark green) and OH (light green) have been highlighted by rectangles above the STY1099 sequence. (**C**) A predicted model of the STY1099 tertiary structure was generated using a publicly available RaptorX platform at www.raptorx.uchicago.edu/StructurePrediction/documentation.

**Figure 2 biology-08-00086-f002:**
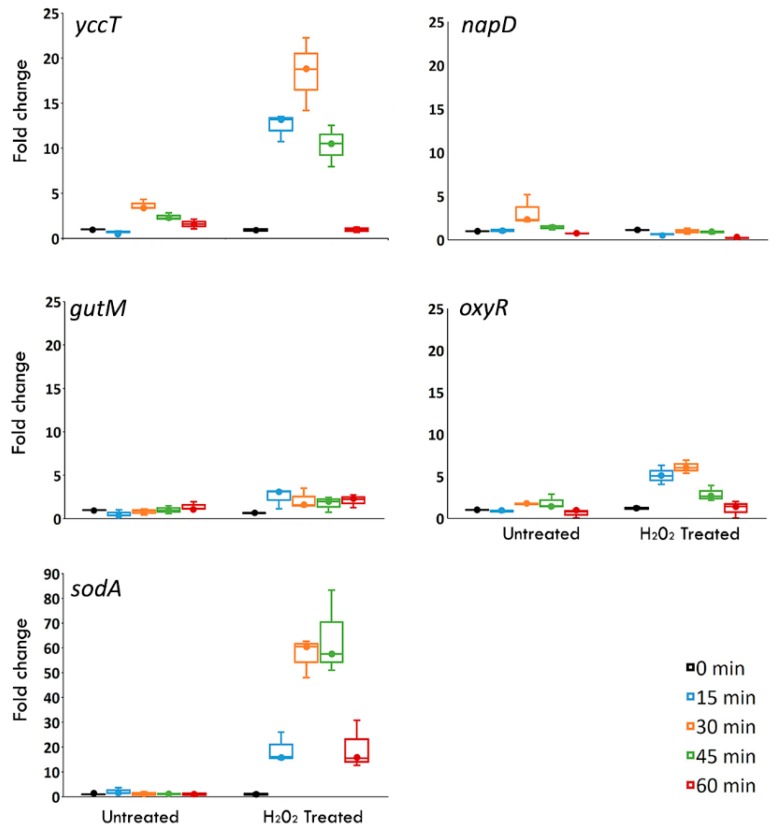
Messenger RNA (mRNA) expression levels of *yccT* (STY1099 protein), *gutM*, *napD* (interactomes of STY1099) and *oxyR*, *sodA* (positive controls for oxidative stress) in the wild type *S. enterica* serovar Enteritidis strain during its exponential growth exposed to 3 mM of H_2_O_2_. Values on the y axis are relative expression levels (fold change) normalized to wild type during the oxidative treatment. The data correspond to the mean value of three biological replications. Error bars correspond to the standard deviation.

**Figure 3 biology-08-00086-f003:**
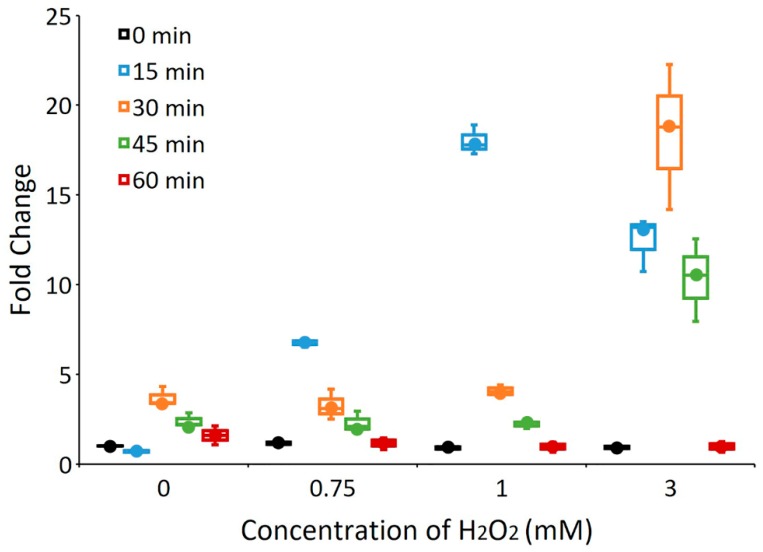
mRNA expression levels of *yccT* in the wild type strain exposed to 0.75 mM, 1 mM and 3 mM concentrations of H_2_O_2_, respectively, during the exponential growth phase. The data correspond to the mean value of three biological replications.

**Figure 4 biology-08-00086-f004:**
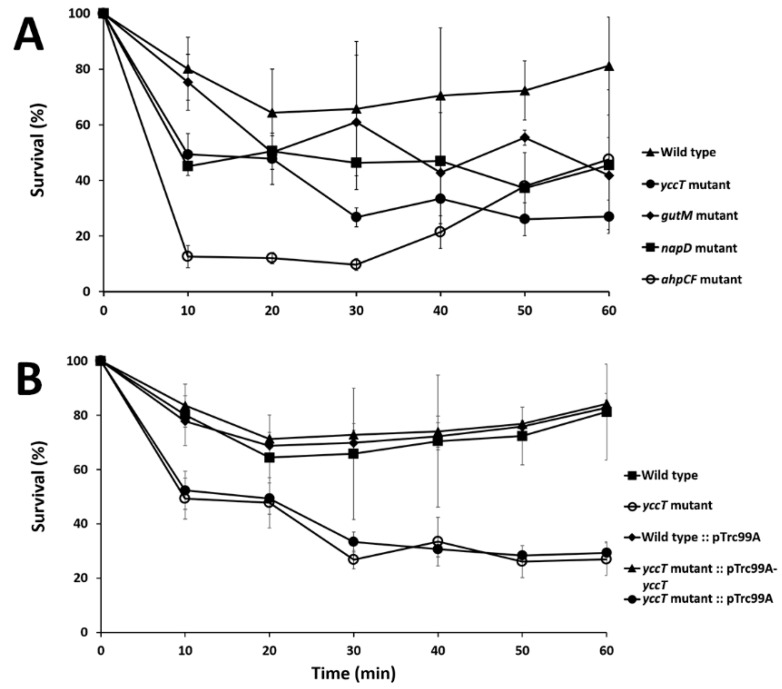
Hydrogen peroxide killing assay. (**A**) Evaluation of the effect of H_2_O_2_ on the ∆*yccT*, ∆*gutM*, ∆*napD* mutant strains compared to that of their parental wild type strain during their exponential growth phases. (**B**) Validation of the effect of the *yccT* gene deletion on the survivability of *S*. Enteritidis during peroxide treatment.

**Figure 5 biology-08-00086-f005:**
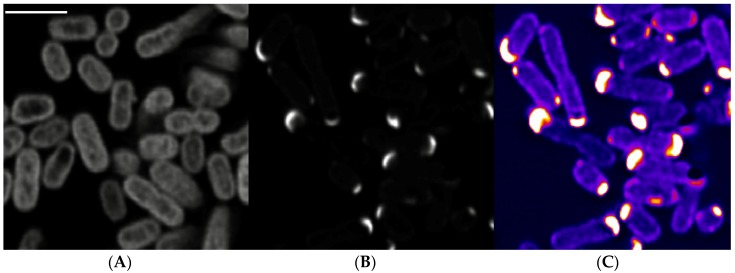
Subcellular localization of STY1099 protein. Single confocal section of deconvolved three D acquisitions are shown. (**A**) GFP expressing bacteria. (**B**,**C**) STY1099-GFP fusions. Panels (**A**,**B**) share the same lookup table, panel C shows the STY1099-GFP fusion with a lookup table intended to show the bacterial cell outline. Scale bar 2 µm.

**Figure 6 biology-08-00086-f006:**
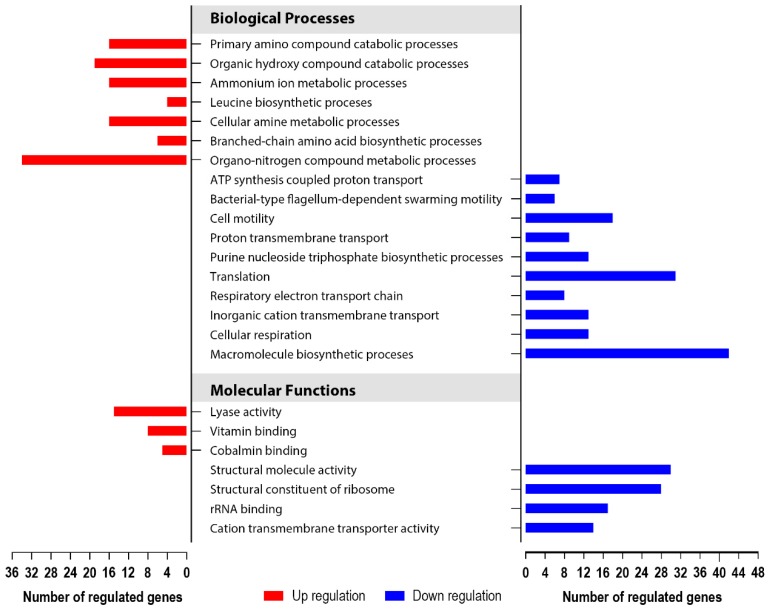
Gene ontology (GO) enrichment analysis portraying the most important biological processes and molecular functions of *S. enterica* serovar Enteritidis during the exponential growth phase that were affected by ∆*yccT* mutation. Included in analysis were genes that were more than two-fold up- or downregulated with FDR (*p* < 0.05) and high reproducibility across the biological replicates.

**Figure 7 biology-08-00086-f007:**
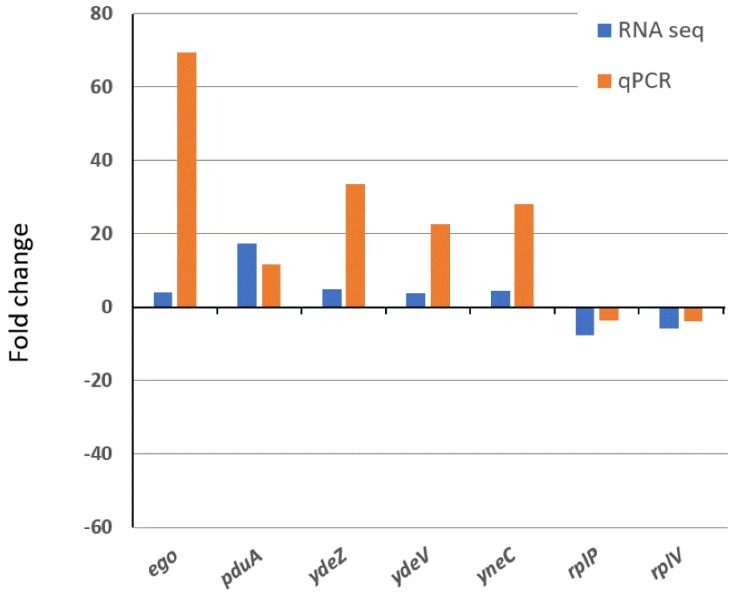
Validation of RNA-sequencing (RNA-seq) data by qRT-PCR analysis. Data represent fold changes in expression of selected seven genes in the wild type and the ∆*yccT* mutant treated with 3 mM H_2_O_2_. Genes differently expressed between the wild type and ∆*yccT* mutant during H_2_O_2_ treatment represent the mean value of three biological replications.

**Figure 8 biology-08-00086-f008:**
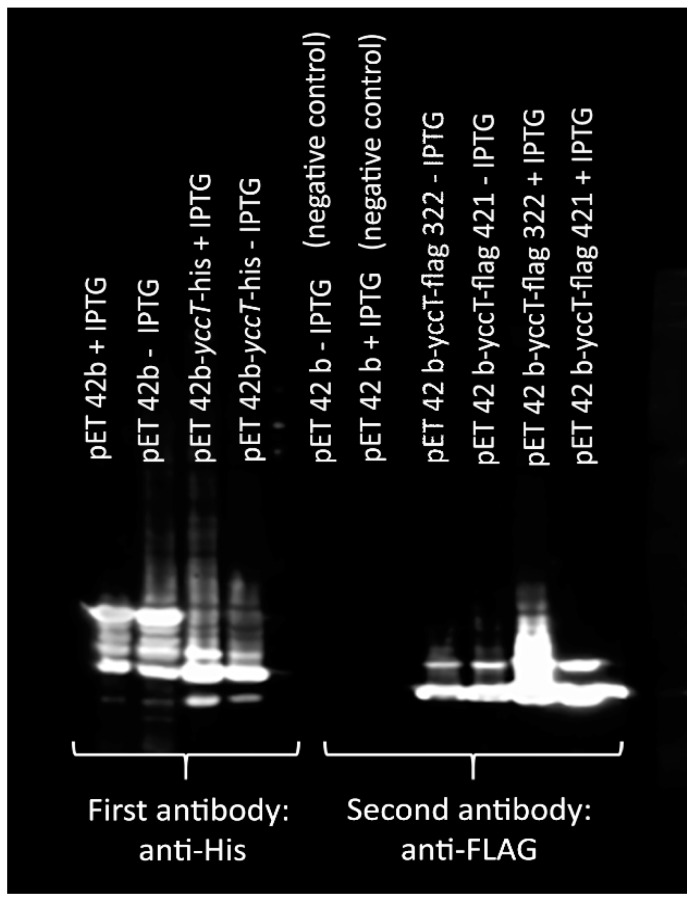
Validation of STY1099 expression of the *yccT*-flag gene. Western blot analysis shows the results for first antibody: anti His (lanes 1–4) and second antibody: anti-FLAG (lanes 5–10). Negative controls lanes 5 and 6.

**Table 1 biology-08-00086-t001:** Primers used in this study for gene deletions and complementation.

Primer Name	Sequence (5′–3′)
**Primers for *yccT*, *napD*, *gutM*, *ahpC* and *ahpF* Deletion**	
*yccT*: Forward	CTT CCT TGC TCT CTG TTT GCC GGT GAC TGT TTT TGC CAC AAC GCT CCG TTT GTA GGC TGG AGC TGC TTC G
*yccT*: Reverse	CCC ATT GCA GAA AAT GAT GGC GTG TCT GCG GGT CGG CCA GTC GGA ACC AAC ATA TGA ATA TCC TCC TTA G
*napD*: Forward	ATG CGC ACT AAC TGG CAG GTC TGT AGC CTG GTC GTG CAG GCC AAA AGT CAT GTA GGC TGG AGC TGC TTC G
*napD*: Reverse	TCA TGG TGT TTC CTC ACC TTG CTC ATC CTG CTG GTG ATA AAC CAG CGA CAC ATA TGA ATA TCC TCC TTA G
*gutM*: Forward	ATG GTT TCC ACT CTG ATT ACC GTC GCC GTT ATC GCC TGG TGT GCG CAA CTT GTA GGC TGG AGC TGC TTC G
*gutM*: Reverse	TTA CCC ATG TTT CAG TTT AAG CGC CAA TGA TAG TGC ATT CTG CGC GAG CGC ATA TGA ATA TCC TCC TTA G
*ahpC*: Forward	ATG TCC TTA ATT AAC ACC AAA ATC AAA CCT TTC AAA AAC CAG GCG TTC AAT GTA GGC TGG AGC TGC TTC G
*ahpC*: Reverse	TTA GAT TTT ACC GAC CAG GTC TAA AGA TGG AGC CAG AGT CGC TTC GCC TTC ATA TGA ATA TCC TCC TTA G
*ahpF*: Forward	ATG CTC GAC ACA AAT ATG AAA ACC CAG CTC AGG GCT TAC CTT GAG AAA CTT GTA GGC TGG AGC TGC TTC G
*ahpF*: Reverse	TTA TGC GAT TTT GGT ACG AAT CAG ATA ATC AAA GGC GCT CAA CGA GGC TTC ATA TGA ATA TCC TCC TTA G
**Primers for Conformation of *yccT*, *napD*, *gutM, ahpC* and *ahpF gene* Deletions**	
*yccT* mutant F	AAA GTG TAC GAC AAA CCT GAC A
*yccT* mutant R	TGA GCA ACC AAC GCG ATA TGT T
*napD* mutant F	GCT GCC AGG ACA GTT GTG AAC C
*napD* mutant R	CCG TTC CGC AAA AAC GGC ACG G
*gutM* mutant F	TTT ATG CCA GCC CGA AAG CGT C
*gutM* mutant R	GCA TGT GCG TTG GCT TTC CTC G
*ahpC* mutant F	ACT TTA GAT GGC TGA CAG GGC GCA
*ahpC* mutant R	CGC GGG TGC GCC CAT GAC TGA AAC
*ahpF* mutant F	GCC GCC TTA CTC TGA CGT GAA ATA
*ahpF* mutant R	TTA ACA GAC CGT TTC AGA GTA TTG
**Primers for *yccT* Mutant Complementation ^a^**	
*yccT*-F	TCA CCC GGG ATT CGA TTC CGT AGT TAA CCT
*yccT*-R	GCG GTC GAC AGC AAA ACG CCA AAA TAT GTC

^a^ Underline bases represent restriction endonuclease sites: SmaI and SalI.

**Table 2 biology-08-00086-t002:** Altered genes during peroxide treatment of the ∆*yccT* mutant of *S.* Enteritidis.

Biological Processes and Genes	Locus Tag	Protein Description	Fold Change	FDR *p*-Value ^1^
Citrate metabolism				
*citE*	SEN0591	Citryl-CoA lyase	2.67	0.0018
*citF*	SEN0590	Citrate CoA-transferase	4.18	0.0307
*citT*	SEN0587	Citrate carrier	2.55	0.0413
Oxidoreductase activity/electron transport				
*hycD*	SEN2692	Respiratory-chain NADH dehydrogenase	2.35	0.0274
*hycF*	SEN2690	Formate hydrogenlyase complex iron-sulfur	2.64	0.0274
*dmsA3*	SEN1552	Dimethyl sulfoxide reductase	3.87	0.0274
SEN1249	SEN1249	Hydrogenases b-type cytochrome	2.12	0.0448
SEN3184	SEN3184	Na^+^-transporting methylmalonyl-CoA/oxaloacetate decarboxylase	2.06	0.0317
rRNA processing/stress response				
SEN0992	SEN0992	Oligogalacturonate-specific porin	4.66	0.0274
*rrmJ*	SEN3130	23S rRNA methyltransferase J	2.12	0.0378
Gluconate transmembrane transport				
*kdgT*	SEN0166	2-keto-3-deoxygluconate permease	5.33	0.0045
Nitrate metabolism				
*narJ*	SEN1277	Nitrate reductase	−16.96	0.0162
*narH*	SEN1276	4Fe–4S ferredoxin, respiratory nitrate reductase	−15.31	0.0178
*narI*	SEN1278	Nitrate reductase, gamma subunit	−11.24	0.0178
*narG*	SEN1275	Respiratory nitrate reductase, subunit alpha	−10.98	0.0162
*narK*	SEN1274	Nitrite extrusion protein	−7.48	0.0162
*nirC*	SEN3303	Nitrite transporter	−3.58	0.0274
Unknown				
SEN0167	SEN0167	Hypothetical protein	4.75	0.0274
SEN0271	SEN0271	Hypothetical protein	2.33	0.0306
*yycT* ^2^	SEN0942	STY1099	−1525.68	9.0780 × 10^−56^
SEN1163	SEN1163	Phage membrane protein	−8.98	0.0008
SEN0541	SEN0541	Protein of unknown function DUF1471	−2.31	0.0274

^1^ False discovery rate (FDR) *p* < 0.05; ^2^ Deleted *yccT* gene that encodes STY1099 protein.

**Table 3 biology-08-00086-t003:** Proteins of *S. enterica* Enteritidis that form an interactom with the STY1099 protein.

Biological Processes and Protein Names	Accession Number	Identification Probability (%)	Molecular Weight	Identified Unique Peptide Sequences
Cell Division and Shape				
Septum site-determining protein MinD	YP_005216705.1	100	29,509.3	ASNQGEPVILDATADAGK; AYADTVDR; IIVVTSGK; IKLVGVIPEDQSVLR; LVGVIPEDQSVLR; TENLFILPASQTR; TTSSAAIATGLAQK
Cell division protein FtsH	ZP_12129674.1	100	70,784.5	QKLESQISTLYGGR; QVVVGLPDVR
Rod shape-determining protein MreB	ZP_11739057.1	100	36,934.4	DGVIADFFVTEK; GMVLTGGGALLR; IKHEIGSA YPGDEVR; IKHEIGSAYPGD EVREIEVR; NLAEG VPR; NYGSLIGEATAER; RNYGSLIGEATAER; VL VCVPVGATQVER
Oxidoreductase Activity/Electron Transport				
Biotin carboxylase of acetyl-CoA carboxylase	ZP_12119137.1	97.0		VVEEAPAPGITPELRR; YLENPR
Multifunctional fatty acid oxidation complex subunit alpha	ZP_15867656.1	100	79,596.2	DFSDDEIIAR; KEEDAAVDDLLASVSQTKR; QAI TGDLDWR
Glutamate dehydrogenase	ZP_15809056.1	100	45,978.8	CAALNLPYGGAK; LFAGAGAR; RANIAVEGAR; TAAYIVACER; VAVQGFGNVGSEAAR
Precorrin-4 C11-methyltransferase	ZP_11737843.1	100	28,358.5	EQGEELTR; GTLADISDKVR; LQTGDVSLYGSVR
Short chain dehydrogenase	ZP_13073001.1	100	27,852.2	ADVRDFASVQAAVAR; AKETEGRIDILVNNAG VCR; SLAVEYAQSGIR; TALITGASQGIGEGIAR; TPMAESIAR; VNAICPGYVR
Aminoacyl-histidine dipeptidase	YP_005235990.1	100	52,419.9	EAVPAGFACFK; FLAGHAEELDLR; LLNATPNGVIR
Cytochrome d terminal oxidase subunit 1	ZP_15859114.1	99.9	58,265.5	AYELLEQLR; DLGYGLLLKR
Anaerobic glycerol-3-phosphate dehydrogenase subunit A	YP_005397674.1	100	57,893.3	ACEAAGIR; AEAIDPQQAR; EGATVCGVHVR; H DIATGATGR; HGDRTPGWLSEGR; IAEYADLSI R; IISLPAPLR; INQHVINR
Bifunctional acetaldehyde-CoA/alcohol dehydrogenase	ZP_15844816.1	96.9	96,199.8	AVTNVAELNALVER; AAALAAADAR
Transcription and Translation				
30S ribosomal protein S3	ZP_15868039.1	100	25,965.5	EFADNLDSDFKVR; EGRVPLHTLR; GIKVEVSGR; IVIERPAK; KPELDAK; KVEVSGR; KVVADIAGPAQINIAEVR; LGGAEIAR; LVADSITSQLER; VPLH TLR
DNA-binding transcriptional regulator PhoP	ZP_15835390.1	99.7	25,483.5	IQAQYPHDVITTVR; VLVVEDNALLR
30S ribosomal protein S9	YP_002638940.1	100	14,790.5	AENQYYGTGR; GGGISGQAGAIR; SLEQYFGR
30S ribosomal protein S11	NP_462321.1	100	13,812.8	ALNAAGFR; CADAVKEYGIK; STPFAAQVAAER
LSU ribosomal protein L6p	ZP_12153653.1	100	18,841.3	APVVVPAGVDVK; DGYADGWAQAGTAR; GA DKQVIGQVAADLR; KLQLVGVGYR; YADEVVR
30S ribosomal protein S4	ZP_09771019.1	99.6	23,467.7	AALELAEQR; LSDYGVQLR; VKAALELAEQR
Threonyl-tRNA synthetase	YP_002226745.1	100	73,887.2	ALNAYLQR; IYGTAWADKK; LSASYVGEDNER; PVITLPDGSQR
50S ribosomal protein L5	YP_005183280.1	100	20,300.6	GLDITITTTAK; ITLNMGVGEAIADKK; LITIAVPR; QGYPIGCK
cAMP-regulatory protein	ZP_09770968.1	100	23,595.9	QEIGQIVGCSR; VGNLAFLDVTGR
50S ribosomal protein L21	NP_457683.1	95.7	11,560	MYAVFQSGGK; VSEGQTVR
Stress Response				
ATP-dependent protease ATP-binding subunit ClpX	ZP_15869131.1	100	46,159.1	KHPQQEFLQVDTSK; LLYCSFCGK; SNILLIGPT GSGK
Two-component response regulator OmpR	YP_005215137.1	99.8	27,317.4	SIDVQISR; SVANAEQMDR
DNA-binding ATP-dependent protease La	YP_002225562.1	100	87,395.7	DIHVHVPEGATPK; LGINPDFYEKR
DnaK protein (heat shock protein 70)	YP_005211303.1	99.9	69,241.2	FQDEEVQR; RFQDEEVQR
Carbohydrate Metabolism				
Phosphoenolpyruvate synthase	ZP_15868324.1	100	87,132.5	AAAIVTNR; AVIEELAR; IEDVPQQQR
Dihydrolipoamide succinyltransferase	ZP_15861596.1	100	43,825.8	APAVEPAAQPALGAR; LLAEHNLEASAIKGTGV GGR; QYGEVFEKR
Phosphoenolpyruvate carboxylase	YP_005214624.1	100	99,020.2	GEAASNPEVIAR; GGAPAHAALLSQPPGSLK; H VLLLSR
Carbohydrate Metabolism Continued				
S-adenosylmethionine synthetase	ZP_09770040.1	100	42,437.9	HGGGAFSGKDPSKVDR; KIIVDTYGGMAR; QSPDINQGVDR; TDKAQLLR
Amino Acid Metabolism				
Carbamate kinase	ZP_11788746.1	100	33,332.7	DHLVICNGGGGVPVVEK; KNIELAAR; NHLPER; RGEPLEADIQRK; RIVENDAIR; TLVVALGGNALLK; VTACAEFVSHCR; YIGPIYDEAQAR
Glutamate-1-semialdehyde aminotransferase	YP_002242369.1	100	45,417.4	ELIPGGVNSPVR; HTLTCTYNDLTSVR; NAVIEA AER; SKSENLYSAAR
Arginine deiminase	ZP_15827530.1	99.4	45,571.6	AGEEHDIFANTLR; LGPTFAADIR
Lipid Metabolism				
3-oxoacyl-(acyl carrier protein) synthase II	ZP_11731244.1	98.6	41,678.9	ASTPLGVGGFGAAR; SVFGDAASR
Cellular Transport				
Maltose/maltodextrin transporter ATP-binding protein	ZP_14274887.1	100	40,780.7	FVAGFIGSPK; MNDIPPAER; MNFLPVK; QQI WLPVESR; RLHQEPGV; TLVAEPR; VAQVGKPL ELYHYPADR; VTATAIEQVQVELPNR
Oligopeptide ABC transporter ATP-binding protein OppF	ZP_15831204.1	100	37,252.3	HAVSCLKVDPL; LYEGETLGVVGESGCGK; VGLLPNLINR

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
