# Peer review of "Transcriptional Profiling and Molecular Characterization of the yccT Mutant Link: A Novel STY1099 Protein with the Peroxide Stress Response and Cell Division of Salmonella enterica Serovar Enteritidis"

_biology, 2019, doi:10.3390/biology8040086_

Round 1
Reviewer 1 Report
The manuscript presents interesting data, however, this article need a minor corrections:
The title should be clarify "Characterization.... Functional? Genetics? Transcriptomic? The aim of the study should be closely connected with conclusion.
Author Response
The manuscript presents interesting data, however, this article need a minor corrections:
Response # 1. Thank you for the comment.
The title should be clarify "Characterization.... Functional? Genetics? Transcriptomic? The aim of the study should be closely connected with conclusion.
Response # 2. We agreed with the reviewer’s comment. The previous title “Characterization of novel STY1099 protein involved in the peroxide stress response of Salmonella Enteritidis” was general and did not provide sufficient information about the main findings of this study. Therefore, we modified the title of the manuscript, which now reads: “Transcriptional profiling and molecular characterization of the yccT mutant link a novel STY1099 protein with the peroxide stress response and cell division of Salmonella enterica serovar Enteritidis.” We think that the modified title of this manuscript is more informative for the readers and well supported by our experimental data. Equally important it accurately reflects the aim of the study, as mentioned by reviewer #1.
Reviewer 2 Report
The authors have presented their work in the characterization of a novel protein STY1099 in Salmonella Enteritidis by using various techniques to show this gene is highly induced in the presence of hydrogen peroxide and in its absence cells have increased susceptibility to hydrogen peroxide. Transcriptomics of the WT and deletion strains in the absence and presence of hydrogen peroxide provide insight into the function of STY1099 as does the GFP imaging. There are many convincing results presented, however there are also some significant oversights which need to be addressed in order to make any final conclusions.
STY1099’s protein sequence contains a Sec signal sequence (as identified by SignalP) which would infer that the protein is periplasmic and as such would not bind to or interact with any cytoplasmic proteins. To confirm this, I recommend the authors to characterize the subcellular localization of STY1099. If it is periplasmic then many of the current results/discussion points will be redundant as many of the proteins thought to interact with STY1099 were cytoplasmic or inner membrane anchored proteins that face the cytoplasm. As mentioned below the Co-IP experiment needs to be done again with the relevant controls in order to decipher actual binding partners. This should also provide further insight into binding partners of STY1099.
Below are detailed questions or comments that should also be addressed:
Section 3.1 Phylogeny, conserved domains and structural characteristics of STY1099
-Figure 1 legend contains text at the end that is not related.
-Figure1C please state either in the legend or methods how you generated the predicted model of STY1099 and highlight what the domains are on the protein model.
Section 3.2 The ycct gene is highly inducible upon exposure to hydrogen peroxide.
-SodA is used as a control and shows a massive change in expression in the presence of H202. As it is present in the figure, please comment on this result in the text and how it gives context to your results for yccT.
Section 3.4 Subcellular localization of STY1099
-In the methods there is no mention of how GFP was incorporated into the plasmid.
-There is no figure legend for this figure 5. The text present is for Figure 6.
Section 3.5 Global transcriptomics
-I found this section hard to follow. As it stands the H202 treated strains analysis is presented at the start of the section and then revisited at the end of the section on Line360. It would be much easier to follow if these sections were grouped together.
-Ensure the figure legend text matches the figure.
-Line 361. Are the author’s are referring to Table 2 and TableS3 and not Table S2?
Section 3.6 Determination of STY1099
-the Co-IP experiment has no negative controls and as such the current results are not valid. The experiment needs to be performed by using cell lysate (and periplasmic extracts as well would be very informative) from WT and the yccT deletion strain. Showing the SDS-PAGE gel of the proteins also supports the final results.
-The results and final conclusions will need to be revisited once this experiment has been performed.
Author Response
The authors have presented their work in the characterization of a novel protein STY1099 in Salmonella Enteritidis by using various techniques to show this gene is highly induced in the presence of hydrogen peroxide and in its absence cells have increased susceptibility to hydrogen peroxide. Transcriptomics of the WT and deletion strains in the absence and presence of hydrogen peroxide provide insight into the function of STY1099 as does the GFP imaging. There are many convincing results presented, however there are also some significant oversights which need to be addressed in order to make any final conclusions.
Response # 3. Thank you for the comments.
STY1099’s protein sequence contains a Sec signal sequence (as identified by SignalP) which would infer that the protein is periplasmic and as such would not bind to or interact with any cytoplasmic proteins. To confirm this, I recommend the authors to characterize the subcellular localization of STY1099. If it is periplasmic then many of the current results/discussion points will be redundant as many of the proteins thought to interact with STY1099 were cytoplasmic or inner membrane anchored proteins that face the cytoplasm. As mentioned below the Co-IP experiment needs to be done again with the relevant controls in order to decipher actual binding partners. This should also provide further insight into binding partners of STY1099.
Response # 4. After the project completion, the principal investigator of this study changed his affiliation and moved to a new place (New Zealand). Although the reviewer’s recommendation would improve the overall quality of the manuscript, we are currently unable to carry out this experiment.
Regarding the second part of the reviewer’s comment, we carried out two negative controls, during the Co-IP experiment. Our “Material and Methods” were briefly written and we did not indicate information about these negative controls. In this revised version of the manuscript, this information has been included. We also presented several results related to the validity of our Co-IP experiment, including western blot, dot blot, and SDS-PAGE analyses. In this revised version of the manuscript, the readers will have access to this type of data. Also, as pointed by the reviewer on several occasions (for instance, how GFP was incorporated), we paid special care to the “Material and method” section. The current version of the manuscript contains all relevant information in “Material and methods”, so that the reader can properly understand our methodology. We apologize for this inconvenience.
Below are detailed questions or comments that should also be addressed:
Section 3.1 Phylogeny, conserved domains and structural characteristics of STY1099
-Figure 1 legend contains text at the end that is not related.
Response # 5. Figure 1 legend has been fixed.
-Figure1C please state either in the legend or methods how you generated the predicted model of STY1099 and highlight what the domains are on the protein model.
Response # 6. Figure 1 C legend has been modified. The modified version of the figure legend contains the required information. Now it reads: “(C) Predicted model of the STY1099 tertiary structure was generated using a publicly available RaptorX platform at www. raptorx.uchicago.edu/StructurePrediction/documentation.”
Regarding the second part of the reviewer’s comment, the RaptorX software besides a tertiary structure prediction can provide binding sites of a protein but not domains of the protein. In this particular case, we provided an alignment with color coded (e.g. dark blue) indication of conserved regions of STY1099 protein across several bacterial species that may indicate domains of this novel protein.
Section 3.2 The ycct gene is highly inducible upon exposure to hydrogen peroxide.
-SodA is used as a control and shows a massive change in expression in the presence of H202. As it is present in the figure, please comment on this result in the text and how it gives context to your results for yccT.
Response # 7. Thank you for this interesting question. Yes, we used two positive controls, sodA and oxyR, during the gene expression assay. Although both of these genes encode proteins that are hallmarks of the oxidative stress response of Salmonella enterica, we observed differential expression of these two genes, a massive expression of sodA, and much lower expression of oxyR. As suggested, we commented on this result and provided a possible explanation for this difference and gave a context for yccT in a current version of the manuscript.
This new paragraph reads: “Interestingly, another positive control, sodA, showed a massive change in expression in the presence of H2O2, resulting in three and six times higher expression compared to those of yccT and oxyR, respectively, during 30 and 45 min treatments (Fig. 2). A reason of the massive expression of sodA most likely lies in the fact that this gene encodes an enzyme, SodA, which catalyzes an oxidative agent, whereas oxyR encodes a protein that regulates expression of other oxidative stress response genes. In other words, sodA must be expressed at much higher levels compared to that of oxyR, as these two proteins have quite different mode of action combating an oxidative assault. Based on the expression level, the yccT gene is positioned between these two oxidative stress response genes, oxyR and sodA. This may indicate that yccT has a different mode of action during oxidative stress response of S. enterica compared to those of oxyR and sodA.
Section 3.4 Subcellular localization of STY1099
-In the methods there is no mention of how GFP was incorporated into the plasmid.
Response # 8. In the previous version of the manuscript, we provided a brief statement about the fusion of yccT and GFP. Thank you for pointing at this. The description of the methodology was too brief that the readers could not be able to perform the same procedure looking at our methodology. Therefore, in the current version of this manuscript, the procedure of making a fusion of yccT and GFP was explained in detail. We think the current methodology will be sufficient for a reader to repeat the same procedure.
Now this part of “Material and methods” reads: “Subcellular localization of the STY1099-fused with green fluorescent protein (GFP) was carried out in the transformed S. enterica Enteritidis ATCC 13076 wild type strain by inverted TiE deconvolution microscope. Briefly, the yccT-egfp fusion gene was amplified using the genome of S. enterica serovar Enteritidis ATCC 13076 and egfp plasmid (AddGene, USA) as the DNA template. The primers for yccT PCR amplification were 5’– GGAATTCCATATGAAAACCGGCGCGCTAGCCACCTT-3’ and 5’-TCCTCGCCCTTGCTCACCATGGATCCAGAGGGCGGCTGTTTTTCCGC-3’and the primers for egfp PCR amplification were 5’- GCGGAAAAACAGCCGCCCTCTGGATCCATGGTGAGCAAGGGCGAGGA-3’ and 5’- GGAATTCCATATGAAAACCGGCGCGCTAGCCACCTT-3’. Fusion of yccT and egfp was carried using amplified yccT and egfp as DNA templets and primers 5’- GGAATTCCATATGAAAACCGGCGCGCTAGCCACCTT-3’ and 5’- GGAATTCCATATGAAAACCGGCGCGCTAGCCACCTT-3’. The amplified yccT-egfp gene was digested with NdeI and XholI and ligated into the pET42b vector. An egfp gene intended for non-fusion was amplified with 5’-GGAATTCCATATGGTGAGCAAGGGCGAGGA-3’ and 5’- CCGCTCGAGTTACTTGTACAGCTCGTCCA-3’. The amplified egfp gene was digested with NdeI and XhoIl and followed ligation into the pET4b vector. After plasmid isolation and sequence verification, the pET42b::yccT-egfp and pET42::egfp plasmids were used to transform S. Enteritidis ATCC13076 wild type. Concentration of 0.5 mM of IPTG (4’6-diamidino-2-phenylindole) was used to induce expression of the yccTEGFP and EGFP, respectively. Transformed S. Enteritidis cells were spotted on slides with ProLong® Diamond Antifade Mountant (4’6-diamidino-2-phenylindole) and imaged. Images were acquired in a Nikon TiE microscope equipped with a confocal A1R scan head. All images were captured with a 100x Apo TIRF 1.49 NA objective. GFP fluorescence was excited with a 488 nm laser and fluorescence collected through a 500-550 nm emission filter. The confocal aperture was set to 16.6 µm (0.3 AU) and the voxel size was 31x31x100 nm (XYZ). All acquisition parameters were kept constant between experimental and control samples. The images were processed with 3D automatic deconvolution (Nikon Elements software, 20 iterations).”
-There is no figure legend for this figure 5. The text present is for Figure 6.
Response # 9. It was a mistake introduced during the manuscript formatting process. The mistake is corrected and a correct figure legend has been added.
Section 3.5 Global transcriptomics
-I found this section hard to follow. As it stands the H202 treated strains analysis is presented at the start of the section and then revisited at the end of the section on Line360. It would be much easier to follow if these sections were grouped together.
Response # 10. Agreed. The current version of the manuscript contains a single paragraph that describes H2O2 treatment of the wild type and yccT mutant and then focuses on the influence of the yccT gene during this treatment. After this, we presented the data that show an influence of the yccT gene on the exponentially growing Salmonella enterica culture without any H2O2 treatment. At the end of transcriptomic section, we presented the validation of RNA seq data by qRT-PCR. This revised section is more comprehensive compared to the previous version and we think that the readers will find this section easy to follow.
-Ensure the figure legend text matches the figure.
Response # 11. The figures 6 and 7 legends have been corrected. Now in the revised version of the manuscript, all figures legends match their figures.
-Line 361. Are the author’s are referring to Table 2 and TableS3 and not Table S2?
Response # 12. Yes, that is correct. This mistake has been corrected.
Section 3.6 Determination of STY1099
-the Co-IP experiment has no negative controls and as such the current results are not valid. The experiment needs to be performed by using cell lysate (and periplasmic extracts as well would be very informative) from WT and the yccT deletion strain. Showing the SDS-PAGE gel of the proteins also supports the final results.
Response # 13. As stated in our response # 4, we carried out negative controls, but these negative controls have not been included in the previous version of the manuscript. In this version of the manuscript, we included statements relevant to negative controls and the validity of our Co-IP experiment in sections “Material and methods” and “Determination of STY1099 interactome”.
It is important to emphasize that the negative controls and various modifications of real samples were tested by western blot and dot blot. As indicated by the western blot analysis both negative controls (lanes #5 and 6) show no expression of the targeted protein, whereas the real samples PET 42 b-yccT-flag without IPTG (lane # 7) and PET 42 b-yccT-flag with IPTG showed expression of the targeted protein. The western blot indicates that the expression of the targeted protein occurs even without the addition of IPTG although at a much lower level compared to that with the addition of IPTG, which was expected. In addition to western blot, we provided similar results using dot blot analysis. At the end, we run SDS-PAGE gel and checked the eluted mixture of proteins. All these analyses are now presented in the supplementary material (Figures S1, S2 and S3) and can be accessed by the reader.
In addition to these analyses, our Co-iP results have been confirmed by another experimental approach, “Subcellular localization of STY1099”. This analysis clearly showed mono-polar localization of the STY1099 protein, while the Co-iP analysis showed that STY1099 interacts with the key division proteins, septum site-determining protein MinD, cell division protein FtsH and rod shape-determining protein MreB. The location of this protein and its interactome correlate to each other and gave another line of confidence in our Co-iP data.
-The results and final conclusions will need to be revisited once this experiment has been performed.
Response # 14. As mentioned in our responses # 4 and 13, we provided all required data, including western blot, dot blot and SDS-PAGE gel analyses, which indicate the high validity of our Co-iP results.
It is very important to mention that our conclusions are based on multiple lines of evidence. For instance, the involvement of the yccT gene or its product STY1099 protein in the oxidative stress response has been documented by “Gene expression assay, “Oxidative killing assay”, “Complementation study”, “Global transcriptomics” and “Co-iP assay”. A possible role of STY1099 in the cell division is supported by “Co-iP assay” and “Subcellular localization of STY1099 assay” too. Equally important, all of the methods employed in this study contained various controls (negative and positive) that always indicated the high validity of our data.
Round 2
Reviewer 2 Report
I have read the additional information sent to me for the following manuscript
Title: Characterization of novel STY1099 protein involved in the peroxide
stress response of Salmonella Enteritidis
Authors: Sinisa Vidovic *, Xiaoying Liu, Ran An, Kristelle Mendoza, Juan
Abrahante, Anup Johny, Kent Reed
All of my comments/requests were responded to and I thank the authors however, my main two issues still remain contentious.
Firstly, the author’s addressed the problem of no controls presented for the co-IP experiment by adding in changes to the text , with three additional results in the supp section. I still can’t tell if the proteomics was performed on both the control and protein expressed samples and then the different proteins between them reported? A final statement on this in the methods make it so much easier to understand. That is, ecoli lysate alone (lane6) subtracted from Lane7 and 8 samples of pET 42 b-yccT-FLAG in BL21 + S. enterica 541 serovar Enteritidis? They talk about cutting out bands from the SDS-PAGE gel but from the supp figure, I can't see any other proteins except for yccT. The fact they pulled-down ribosomal sunbunits makes me think there is likely a lot of non-specific binding going on even though I'm sure they did catch some really interesting binding partners. Supp Figures should also contain appropriate labelling like markers, protein names etc. and presented like a figure in the main text.
Secondly, the author's made a choice not to mention the fact yccT is most likely periplasmic as it contains a periplasmic signal sequence. I appreciate that the authors cannot perform anymore experiments but I feel it is misleading not to include this information, now that they have been made aware of it. The author’s need to at least comment on this. Even if the authors have other results that support their conclusions, they still need to address the fact it has a periplasmic signal sequence in their discussion at least.
I appreciate that they have fixed up a lot of things, but I would like some final clarity about how the final co-IP list was obtained (was it substrated from the controls or not?) and some form of discussion point about the localization of yccT is very likely periplasmic and putting that in context with their own results.
Author Response
Response # 1.
Yes, lane 6 served as a negative control. This particular negative control was treated with IPTG, while another negative control (lane 5) was carried out without any addition of IPTG. In both negative controls we see no bands indicating no expression of STY1099 protein. Then we run the positive controls (lanes 7-10) to make sure that our ycct gene can be expressed and translated into the STY1099 protein that will serve as a bite for all other proteins that can interact with it. Again we used two types of positive controls, with addition and without addition of inducer IPTG. On the western blot gel can be seen that even without addition of IPTG the ycct gene can be expressed and translated into the STY1099 protein (lanes 7 and 8). With addition of IPTG this expression is significantly greater, which we expected to see. For our Co-IP experiment we used IPTG to make sure that there is enough of STY1099 protein, so that we get enough of these interactions with other proteins. Similar results have been seen using dot blot. We also experimented with different volumes and binding conditions (e.g., using unbinding cell lysates). These results are presented by a SDS-PAGE gel.
The final identification of the STY1099 interactom was performed in the proteomic facility located at the University of Minnesota. They run a SDS-PAGE gel and excised a portion of a gel followed by protein digestion and analysis by LC-MS. During our consultation, I asked the staff at the proteomic facility to send us a SDS-PAGE gel image with all bands and they responded that there will be a smear instead of bands and it would not be of any importance for the readers. So we do not possess a final SDS-PAGE gel.
Regarding the images, we initially wanted to include two images, a western blot gel and a SDS-PAGE gel as a main figure in the text. After this attempt, we realized that the SDS-PAGE gel image is of a very low quality. So we finally prepared the western blot image as a figure for the main text, while the SDS-PAGE was improved as much as possible and deposited as a supplementary figure. We hope with these modifications that the readers will have access to all necessary data to fully understand our experimental procedures.
Secondly, the author's made a choice not to mention the fact yccT is most likely periplasmic as it contains a periplasmic signal sequence. I appreciate that the authors cannot perform anymore experiments but I feel it is misleading not to include this information, now that they have been made aware of it. The author’s need to at least comment on this. Even if the authors have other results that support their conclusions, they still need to address the fact it has a periplasmic signal sequence in their discussion at least.
Response # 2.
Agreed. We first carried out an analysis using SignalP 5.0 software version.
The results are reported in the section “Phylogeny, conserved domains and structural characteristics of STY1099 protein” It reads (lines 246-249): “Also the same protein sequence, with a high likelihood (0.9497), contains a Sec/SPI “standard” secretory signal peptide with a cleavage site located between positions 20 and 21, VFA / TT amino acid sequences (probability 0.9324).”
These findings are also discussed (lines 504-508): “In addition, it has been shown with the high likelihood (0.9324) that STY1099 protein contains a Sec / SPI signal peptide. This secretory signal peptide, transported by the Sec translocon and cleaved by signal peptidase I, is an integral part of periplasmic proteins, which suggests that novel STY1099 protein belongs to a group of periplasmic proteins.”
In conclusion, these new findings are reported in the “Results” section and discussed in “Discussion”. Overall, this new addition nicely correlates with our other data, which makes this manuscript more comprehensive and a better article in general. We thank the reviewer for bringing up this issue, as we did not have an intention not to discuss it, but we were focused on the second part of the comment and somehow forgot to discuss the existence of Sec / SPI signal peptide in STY1099 protein. We think that in this version of the manuscript this mistake is fixed.
I appreciate that they have fixed up a lot of things, but I would like some final clarity about how the final co-IP list was obtained (was it substrated from the controls or not?) and some form of discussion point about the localization of yccT is very likely periplasmic and putting that in context with their own results.
Response #3.
Thank you for these additional questions. It gives us an opportunity to clarify them and to make this manuscript more comprehensive for the readers.